

# Using reduced-complexity volcanic aerosol and climate models to produce large ensemble simulations of Holocene temperature

Magali Verkerk[1], Thomas J. Aubry[1], Chris Smith[2,3], Peter O. Hopcroft[4], Michael Sigl[5], Jessica E. Tierney[6], Kevin Anchukaitis[6], Matthew Osman[7], Anja Schmidt[8,9], Matthew Toohey[10]

[1]Department of Earth and Environmental Sciences, University of Exeter, Penryn, TR10 9FE, UK
[2]Department of Water and Climate, Vrije Universiteit Brussel, Brussels, 1050, Belgium
[3]International Institute for Applied Systems Analysis, Laxenburg, A-2361, Austria
[4]School of Geography, Earth & Environmental Sciences, University of Birmingham, Birmingham, B15 2TT, UK
[5]Climate and Environmental Physics & Oeschger Centre for Climate Change Research, University of Bern, Bern, 3012, Switzerland
[6]Department of Geosciences, The University of Arizona, Tucson AZ 85721 USA
[7]Department of Geography, University of Cambridge, Cambridge, CB2 3EN, UK
[8]Institute of Atmospheric Physics (IPA), German Aerospace Center (DLR), Oberpfaffenhofen, 82234, Germany
[9]Meteorological Institute, Ludwig Maximilian University of Munich, Munich, 80333, Germany
[10]Institute of Space and Atmospheric Studies, University of Saskatchewan, Saskatoon, Saskatchewan, S7N 5E2, Canada

*Correspondence to*: Magali Verkerk (mv393@exeter.ac.uk)

**Abstract.** Volcanic eruptions are one of the most important drivers of climate variability, but climate model simulations typically show stronger surface cooling than proxy-based reconstructions. Uncertainties associated with eruption source parameters, aerosol-climate modelling and internal climate variability might explain those discrepancies but their quantification using complex global climate models is computationally expensive. In this study, we combine a reduced-complexity volcanic aerosol model (EVA_H) and a climate model (FaIR) to simulate global mean surface temperature from 6755 BCE to 1900 CE (8705 to 50 BP) accounting for volcanic forcing, solar irradiance, orbital, ice sheet, greenhouse gases and land-use forcing. The models' negligible computational cost enables us to use a Monte Carlo approach to propagate uncertainties associated with eruption source parameters, aerosol and climate model parameterisations, and internal climate variability. Over the last 9000 years, we obtain a global-mean volcanic forcing of -0.15 W.m⁻² and an associated surface cooling of 0.12 K. For the 14 largest eruptions (injecting more than 20 Tg of $SO_2$) of 1250 CE – 1900 CE, a superposed epoch analysis reveals an excellent agreement on the mean temperature response between our simulations, scaled to Northern Hemisphere summer temperature, and tree ring-based reconstructions. For individual eruptions, discrepancies between the simulated and reconstructed surface temperature response are almost always within uncertainties. At multi-millennial timescales, our simulations reproduce the Holocene global warming trend, but exhibit some discrepancies on centennial to millennial timescales. In particular, the Medieval Climate Anomaly to Little Ice Age transition is weaker in our simulations, and we also do not capture a relatively cool period in climate reanalyses between 3000 BCE and 1000 BCE (5000 and 3000 BP). We discuss how uncertainties in land-use forcing and model limitations might explain these



differences. Our study demonstrates the value of reduced-complexity volcanic aerosol-climate models to simulate climate at annual to multi-millennial timescales.

## 1 Introduction

Volcanic eruptions can inject large amount of sulfur dioxide ($SO_2$) into the stratosphere, where the resulting sulfate aerosols have a lifetime of 1-3 years, scatter shortwave radiation and absorb longwave radiation. Volcanic forcing results in a net surface cooling and drying at the global scale, although it also affects large-scale modes of climate variability (e.g. monsoons) which strongly modulate regional climate responses (Marshall et al., 2022). Volcanic forcing is one of the main drivers of climate variability at annual to multidecadal time scales (Myhre et al., 2014; Sigl et al., 2015), and volcanic climate impacts have been linked to societal impacts such as crop failure and famines (e.g., Oppenheimer, 2015; Manning et al., 2017; Ljungqvist et al., 2024).

Past volcanic impacts on climate are typically investigated using two approaches: proxy-based reconstructions and model simulations. At an annual resolution, proxies include tree ring growth which dominate reconstructions for the last millennium (e.g. PAGES 2k Consortium, 2019). Tree ring records are sparsely situated and seasonally biased (King et al., 2021; Anchukaitis and Smerdon, 2022). Global Circulation Models (GCMs) or Earth System Models (ESMs) simulate past climate based on volcanic emission or forcing datasets (e.g., Timmreck, 2012; van Dijk et al., 2024; Wainman et al., 2024).

The comparison between the reconstructions and the simulations of past temperature shows discrepancies when it comes to the impact of volcanic eruptions (e.g. Mann et al., 2013; Marshall et al., 2024). Potential explanations include an underestimation of the cooling in the proxies (Mann et al., 2012; Anchukaitis et al., 2012; D'Arrigo et al., 2013; Lücke et al., 2019; Anchukaitis and Smerdon, 2022), or biases and uncertainties in estimating the dates, locations and SO2 emissions of past eruptions using ice cores (Marshall et al., 2019; Toohey and Sigl, 2017) and climate modelling uncertainty resulting in an overestimation of the cooling (Zanchettin et al., 2019; Aubry et al., 2020; Chylek et al., 2020; Marshall et al., 2021).

Numerous uncertainties affect the simulation of volcanic impacts on climate: i) uncertainties on eruption source parameters (e.g. $SO_2$ emission mass, latitude, height and season) which govern the forcing (e.g. Marshall et al., 2019); ii) aerosol modelling uncertainties (e.g. for the Tambora 1815 eruption, interactive stratospheric aerosols differ by a factor ~2 on both the timescale and magnitude of aerosol optical properties perturbation in response to the same volcanic $SO_2$ emission (Clyne et al., 2021)); iii) climate modelling uncertainties (i.e. for the same prescribed aerosol optical property perturbation, the radiative forcing and climate response differ significantly (Zanchettin et al., 2022)), and; iv) uncertainties caused by the modulation of the forced response by internal climate variability (e.g. Zanchettin et al., 2019). Propagating all these uncertainties with GCMs or ESMs is currently highly computationally expensive, leading to most uncertainties being ignored. For example, the Paleoclimate Model Intercomparison Project (PMIP) last millennium experiment (Jungclaus et al., 2017) recommends a 10-member ensemble ran with the recommended aerosol optical property dataset (Toohey and Sigl,



2017), hence assessing only the internal variability (iv) and climate modelling (iii) uncertainty subject to sufficient participating models. Even for experiments much shorter than the last millennium (e.g. McConnell et al., 2020; van Dijk et al., 2022), uncertainties on emission parameters (i) and aerosol modelling (ii) are not accounted for. A handful of studies have quantified the impact of emission uncertainties (e.g. Zanchettin et al., 2019) but they do not account for aerosol (ii) or

70 climate modelling (iii) uncertainties.

Given the cost of rigorously quantifying uncertainties on volcanic climate impacts using GCMs—not least those with interactive stratospheric aerosol schemes—reduced-complexity models are required. Hereafter, we refer to "reduced-complexity models" as an idealised models relying on a limited number of parameters allowing one to reproduce key metrics of complex aerosol or climate models with a low computing cost. For example, for volcanic aerosol modelling, the Easy

Volcanic Aerosol model (EVA, Toohey et al., 2016 and its extension EVA_H, Aubry et al., 2020) represents the stratosphere using 3-8 boxes in which the aerosol production, loss and transport are parametrised using constant timescales. EVA has been widely applied during the Coupled Model Intercomparison Project phase 6 (CMIP6, Eyring et al., 2016), for VolMIP (Zanchettin et al., 2022) and PMIP (Jungclaus et al., 2017). EVA_H has been used to propagate uncertainties on emission parameters to estimates of aerosol optical properties (e.g. Mackay et al., 2022; Vernier et al., 2024). Reduced-complexity

climate models are also increasingly used in climate science and have a dedicated Model Intercomparison Project (Nicholls et al., 2020). These models have recently been applied to volcanic eruptions. For example, the FaIR climate model (Smith et al., 2018; Leach et al., 2021) was used to estimate the global mean surface temperature (GMST) response to the 2019 Raikoke eruption (Vernier et al., 2024) or the 2022 Hunga Tonga Hunga Ha'apai eruption (Jenkins et al., 2023), and to include future volcanic scenarios into climate projections (Chim et al., 2024). The aerosol optical properties, radiative

forcing and surface temperature impacts of an eruption can be rapidly assessed with the Volc2clim webtool (Schmidt et al., 2023; Vernier et al., 2024) which combines EVA_H and FaIR. Lücke et al. (2023) recently combined EVA and a reduced-complexity climate model to produce a large ensemble of GMST timeseries for the last millennium. They conclude that accounting for emission uncertainties (timing of eruption and mass of $SO_2$) partly explains the proxy-model discrepancies on volcanic cooling. Their approach could be further enhanced by accounting for uncertainties associated with the latitude and

altitude of volcanic injections, as well as uncertainties on parameters of reduced-complexity aerosol and climate models used.

Here, we combine reduced-complexity volcanic aerosol (EVA_H) and climate (FaIR) models to simulate the GMST response to volcanic eruptions during the last 9000 years, and we show that in general the Holocene surface temperature can be well reconstructed with our method. Section 2 presents forcing datasets, and the proxy reconstructions, model simulations

and data assimilations to which we compare our ensemble simulations. To test whether apparent discrepancies between proxy-based reconstructions and climate model simulations can be reconciled by uncertainties, we use Monte Carlo simulations to propagate uncertainties associated with $SO_2$ emission parameters, volcanic aerosol and climate modelling, and internal variability (Sect. 3). Sections 4 and 5 present and discuss estimates of radiative forcing and temperature response, and their comparison to proxy reconstructions.



## 2 Datasets

### 2.1 Volcanic Stratospheric Sulfur Injection (VSSI)

To simulate stratospheric aerosol optical properties (Sect. 3.1) and associated radiative forcing (Sect. 3.2), we need to
constrain the timing, mass, latitude and altitude of past volcanic $SO_2$ injections. Datasets based on bipolar ice-core records
provide estimates of the year and mass of sulfur injection. Here we use the HolVol (v1.0) dataset, covering the 9500 – 500
BCE period (Sigl et al., 2022), and the eVolv2k (version 3) dataset (Toohey and Sigl, 2017) for the period 500 BCE – 1900
CE as it relies on a larger number of ice-cores.

Constraining the latitude and the altitude of injection requires linking ice-core sulfate deposition to specific known eruptions.
Of the 920 injections recorded between 6755 BCE and 1900 CE (8705 to 50 BP), we identified in the literature 28 eruption
source matches (Table S1). 15 of these are particularly robust as they stem from matching the geochemistry of cryptotephra
deposited in the ice-core with that of proximal eruption deposits (de Silva and Zielinski, 1998; Zdanowicz et al., 1999; Cook
et al., 2018; McConnell et al., 2020; Pearson et al., 2022; Piva et al., 2023; Plunkett et al., 2023). For 1628 BCE Aniakchak
II and 43 BCE Okmok II eruptions, we use the date and mass of Pearson et al. (2022) instead of that in HolVol/eVolv2k as it
is based on a larger set of ice-cores. 13 matches were obtained based on the coincidental occurrence of a known eruption and
a sulfur deposit in the ice-core. We kept coincidental matches only for eruptions after 1750 CE, all consistent with the
CMIP7 historical dataset (Aubry et al., in prep). This results in the removal of some of the matches proposed in eVolv2k
(1873 Grimsvötn, 1831 Babuyan Claro, 1739 Shikotsu, 1721 Katla, 1707 Fujisan, 1673 Gamkonora, 1667 Shikotsu, 1640
Parker, 1595 Nevado del Ruiz, 1585 Colima, 1510 Hekla) and the addition of two new matches (1846 Hekla and 1760 Kie
Besi).

For identified eruptions, we use the latitude of the source volcano. For unidentified eruptions, eVolv2k and HolVol attribute
a single latitude to events with Greenland-only (≈45°N or 48°N, respectively in eVolv2k and HolVol), Antarctica only
(≈45°S or 37°S), or bipolar (≈0° or 5°N) deposition. In HolVol volcanic ice-core signals in Greenland of ≥10 years durations
are attributed to Iceland (64°N). Here we instead scale the eruption latitude with the asymmetry ratio, using the best-fit
relationship based on the CMIP7 historical dataset and geochemical matches prior to 1750 (Aubry et al., in prep) (Eq. (1)):

$$\lambda = 49.28 \times \mathrm{asym} + 5.749 , \tag{1}$$

with $asym = \left(M_{SO_4}^{NH} - M_{SO_4}^{SH}\right)/\left(M_{SO_4}^{NH} + M_{SO_4}^{SH}\right)$ the asymmetry of the deposit, $M_{SO_4}^{NH}$ and $M_{SO_4}^{SH}$ are the mass of $SO_4$ deposited
in Greenland and Antarctica ice-cores respectively, $\lambda$ is the latitude in degrees N. Applying this formula to the 1831 injection
gives a latitude of 44.2°N which corresponds well to the latest proposed match (Zavaritski Caldera, 46.9°N, Hutchison et al.,
in review).



When an eruption is identified, we collect its plume height estimate from the literature whenever available. When the height is derived from isopleth maps (i.e. clast size obtained from deposit survey (Carey and Sparks, 1986; Rossi et al., 2019), it is very likely to overestimate the mean $SO_2$ injection height, so we use the empirical correction of Engwell, Aubry et al. (2023) (Eq. (2)):

$$H_{SO_2} = H_{iso}/1.49 ,\tag{2}$$

with $H_{SO_2}$ corresponding to the $SO_2$ injection height and $H_{iso}$ to the height estimated from isopleths, both expressed in km above vent level. In the vast majority of cases, the injection height is unknown. We choose to scale the injection height on the default mass of $SO_2$ ($M$) listed in our VSSI datasets, following the relationship used in the CMIP7 historical dataset (Aubry et al., in prep) (Eq. (3)):

$$H = 15.61 \times M^{0.1585} ,\tag{3}$$

Figure 1.f provides an overview of the final volcanic $SO_2$ emission parameters we used (provided in supplementary materials) after application of above methods.





**Figure 1: Forcings used in our study; a. Greenhouse gases forcing (CH₄, N₂O, CO₂); b. Net ice sheet forcing; c. Orbital forcing; d. Land-use forcing; e. Solar forcing; f. Spaciotemporal distribution of volcanic stratospheric sulfur injections. All the forcings are global forcing at the top of the atmosphere, expressed with respect to the 1860-1880 mean forcing.**





## 2.2 Other forcings

**Greenhouse gases**

To prescribe the concentrations of major greenhouse gases, we use the Köhler et al. (2017) database of $CH_4$, $CO_2$ and $N_2O$ atmospheric concentrations derived from ice core records. Although concentrations are provided at annual timestep in Köhler et al. (2017), the mean resolution for $CO_2$ concentration measurements is coarser (e.g., 93 years between 7045 BCE and 136 CE). Figure 1.a shows the radiative forcing resulting from GHGs concentration changes, as calculated by FaIR

(Sect. 3.3).

**Forcing due to ice sheets**

The albedo forcing and the longwave radiative forcing due to orographic changes following the retreat of ice sheets (Planck emissivity effect) have been calculated from transient Holocene simulations with the HadCM3-M2.1d climate model (Hopcroft et al., 2023). These simulations used the ICE-6G_C (VM5a) reconstruction of ice sheets and sea level (Argus et

al., 2014; Peltier et al., 2015) (Fig. 1.b).

**Orbital forcing**

Variations of the Earth's orbital parameters are a driver of climate changes, with successive ice ages being driven by precession, obliquity and eccentricity (Hays et al., 1976). Here we use the radiative forcing resulting from changes in the orbit as calculated by Hopcroft et al. (2023) (Fig. 1.c). They simulate the incoming solar radiation at the top of the

atmosphere using HadCM3-M2.1d based on changes in the orbit from Berger (1978). As we use a reduced-complexity climate model, we require only a global annual mean forcing, calculated as the annual-average of the monthly shortwave top-of-atmosphere incoming radiation multiplied by $(1 - a_p)$, with $a_p$ the preindustrial monthly planetary albedo. However, orbital forcing has strong latitudinal and seasonal dependences, both of which our model cannot capture.

**Anthropogenic land-use forcing**

We account for anthropogenic land-use forcing using three different reconstructions (Fig. 1.d): HYDE 3.2 (Klein Goldewijk et al., 2017), its update, HYDE 3.3, that includes archaeological assessments from ArchaeoGLOBE Project (2019) and radiocarbon dates for Eurasia from the Landcover6K project (pers.comm M. van der Linden), and the KK10 dataset (Kaplan et al., 2011). HYDE 3.2 may underestimate the land-use outside of Europe, while KK10 may be too intense outside of Europe, based on comparison to pollen-inferred land cover and archaeological data (ArchaeoGLOBE Project, 2019;

Hopcroft et al., 2023). As HYDE 3.3 presents an intermediate forcing and is the most recent, we use it as our default land-use forcing, but we test the sensitivity of our results to the choice of land-use forcing reconstructions (Sect. 5.2 and 5.3).

**Solar forcing**

We use the SATIRE-M dataset of total irradiance (Wu et al., 2018) which provides a timeseries for 6755 BCE – present with an annual resolution (Fig. 1.e). They use a solar surface model to invert the solar irradiance from sunspot observations (after

1600) or cosmogenic isotopes measurement ($^{14}$C and $^{10}$Be, before 1600).





## 2.3 Temperature reconstructions and assimilations

We compare our simulated model temperature to the following proxy reconstructions and data assimilations:

1) The Last Glacial Maximum Reanalysis (LGMR), which provides spatially resolved surface temperature globally for the past 24000 years with a 200-year timestep (Osman et al., 2021). The LGMR was produced by assimilating proxies for sea surface temperature (e.g. $\delta^{18}O$ of foraminifera into simulations from iCESM1.2 and iCESM1.3; Brady et al., 2019).

2) The Holocene data assimilation of Erb et al. (2022) of 711 proxy records from the Temperature 12k proxy database, including both oceanic and terrestrial records (e.g. $\delta^{18}O$, pollen). Two climate models are used as model prior: a
HadCM3 simulation of the past 23 ka and the TraCE-21k simulation, with a decadal resolution. The prior climate state is updated with the proxy data following a similar approach as in the LGMR.

3) The PAGES 2k reconstruction (PAGES 2k Consortium, 2019), a GMST reconstruction for the last two millennia. It is based on the PAGES2k multiproxy dataset (PAGES2k Consortium, 2017), which includes both marine and continental records (e.g. geochemistry of marine and lake sediments, tree rings).

4) The King et al. (2021) May to August Northern Hemisphere extratropical (30-90°N) (NH MJJA) temperature reconstruction covering the period 750 – 2011 CE. They assimilate 54 tree ring records into ten different model priors.

5) Four tree ring-based reconstructions of the midlatitudinal Northern Hemisphere summer: NTREND2015 (Wilson et al., 2016), Büntgen2021 (Büntgen et al., 2021), Guillet2017 (Guillet et al., 2017), Schneider2015 (Schneider et al.,
2015). These reconstructions were calibrated to represent land only temperatures between 40 and 75°N.

The main limitations of data assimilations are the use of a limited number of climate model priors that do not consider volcanic forcing (LGMR, Holocene data assimilation and PAGES2k) or utilise outdated volcanic forcing reconstructions (King et al., 2021) and both biases linked to the proxy records (e.g. seasonal and spatial sampling) and to the model
simulations propagate to reconstructions. For some global temperature reconstructions (Erb et al., 2022; Kaufman et al., 2020; PAGES 2k Consortium, 2019), mid latitude Northern Hemisphere records dominate the datasets, time resolution of records varies depending on the proxy (e.g., from annual to millennial resolution), and the seasonality recorded by the proxies can also lead to a mean response with a reduced interannual variability (Anchukaitis and Smerdon, 2022) or longer term biases (e.g. Essell et al., 2024).

## 2.4 Model simulations

We also compare our simulations to the following model simulations:

1) TraCE-21k-II (He and Clark, 2022), a simulation of the global temperature over the last 21 000 years with a 10-year timestep, performed with a coupled atmosphere-ocean general circulation model (CCSM3; Yeager et al., 2006). TraCE simulation is forced by greenhouse gases, fresh water forcing before ~ 14.7 ka, orbital and ice sheets
variations.

2) UKESM1 simulations of GMST and 40-75N land only summer (MJJA) temperature from Marshall et al. (2024) between 1250 and 1849 CE, and their comparison with five other simulations that ran the PMIP4 last millennium experiment. Simulations were performed using $SO_2$ emission driven models (UKESM1, CESM2(WACCM6ma)





and MRI-ESM2) or aerosol optical properties driven models (MIROC-ES2L, MPI-ESM1-2-LR and IPSL-CM6A-LR). They used the $SO_2$ emissions provided in the eVolv2k dataset, and aerosol optical properties are derived from eVolv2k using the reduced complexity aerosol model EVA.

3) Reduced complexity climate model simulations of the 885 – 2000 CE temperature from Lücke et al. (2023), accounting for solar, greenhouse gases, anthropogenic aerosols (after 1820) and volcanic forcings. The model is calibrated using simulations from the HadCM3 GCM. They perform two calibrations to simulate: i) global annual mean surface temperature using global annual mean volcanic forcing, and; ii) mid-latitudinal Northern Hemisphere summer land surface temperature using NH annual mean volcanic forcing. The second calibration enables more consistent comparison with tree ring-based temperature reconstructions. Furthermore, they also test sensitivity to volcanic forcing uncertainty by randomly perturbing the mass of $SO_2$ injected and the year and month of eruption. Beyond the different period simulated in our study (6755 BCE – 1900 CE instead of 885 – 2000 CE), key differences between our study and Lücke et al.'s include:

- We use a revised list of identified eruptions (see Sect. 2.1)
- Instead of EVA, we use EVA_H which accounts for the height of injection (see Sect. 3.1)
- We use a different reduced-complexity climate model (FaIR, see Sect. 3.3) which was extensively used in CMIP6
- We use a different approach to estimate Northern Hemisphere temperature (see Sect. 3.3).
- We account for uncertainties in VSSI latitude and altitude (unaccounted for in Lücke et al.) but not in injection timing (accounted for in Lücke et al.)
- We account for uncertainties in aerosol forcing modelling, climate modelling and internal climate variability (unaccounted for in Lücke et al.) (see Sect. 3.4)

## 3 Models and uncertainty quantification

To simulate the 6755 BCE – 1900 CE (8705 to 50 BP) surface temperature, we first use the reduced-complexity volcanic aerosol model EVA_H (Sect. 3.1) and a volcanic forcing scaling (Sect. 3.2) to estimate volcanic radiative forcing from our emission dataset (Sect. 2.1). We then input the volcanic forcing and all other forcings (Sect. 2.2) into the reduced-complexity climate model FaIR to simulate temperature evolution (Sect. 3.3). Uncertainties in volcanic emission, aerosol and climate model parameters and internal variability are propagated using Monte Carlo simulation (Sect. 3.4). Figure 2 illustrates our full modelling workflow.



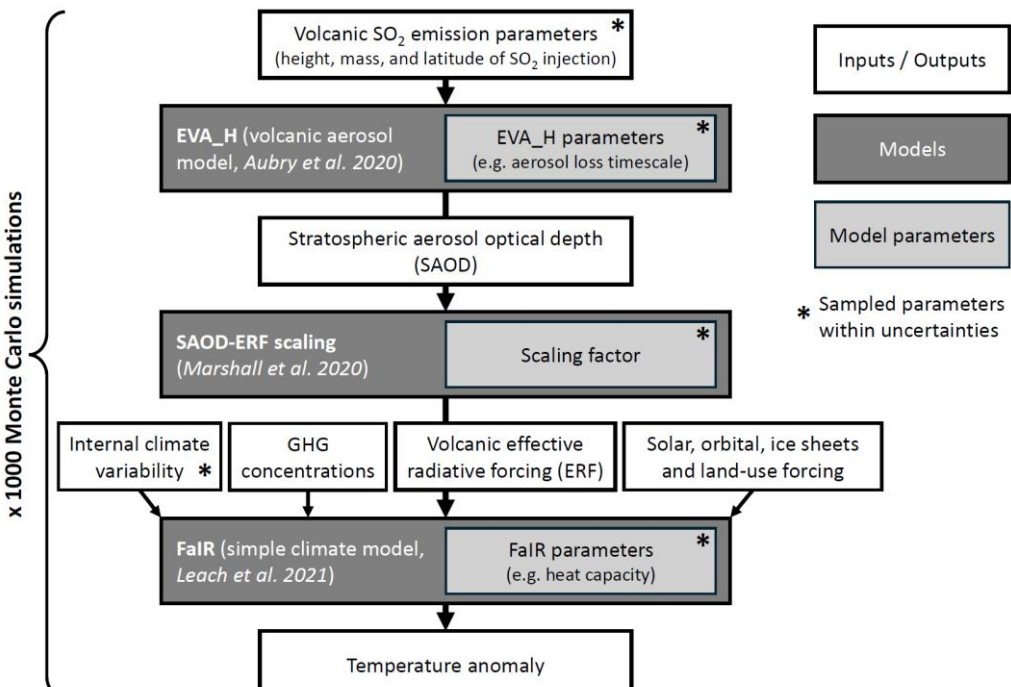

**Figure 2: Flowchart of the modelling process.**


## 3.1 Stratospheric volcanic aerosol model: EVA_H

EVA_H is a reduced-complexity stratospheric aerosol model in which the stratosphere is represented by eight boxes corresponding to different vertical and latitudinal regions (Aubry et al., 2020). Volcanic $SO_2$ injections are converted to sulfate aerosols (aerosol production) and transported horizontally (aerosol mixing) and downward (aerosol loss) between boxes. The key inputs are the mass of $SO_2$, the latitude, height, and date of injection, provided by the databases described in Sect. 2.1. The key outputs are 4D (time, wavelength, latitude and altitude) stratospheric aerosol optical properties from which the global mean Stratospheric Aerosol Optical Depth (SAOD) at wavelength 550 nm can be calculated.

In contrast to its predecessor model EVA (Toohey et al., 2016) used in Lücke et al. (2023), EVA_H's account for the $SO_2$ injection altitude. The injection latitude affects both the latitudinal distribution of SAOD and its global mean in EVA_H, whereas it only affects the former in EVA. Furthermore, EVA was calibrated against the Pinatubo 1991 eruption only, whereas EVA_H was calibrated using the full 1979-2016 satellite observations spanning a large parameter space in terms of volcanic injection magnitude, latitude and altitude. In addition to a best estimate of EVA_H parameters (e.g. aerosol production, loss, and mixing timescales), an ensemble of 100 parameter sets was provided by repeating the calibration procedure with resampling of the input $SO_2$ injection parameters and target aerosol optical properties within their



uncertainties. Given EVA_H's empirical nature, results should be interpreted cautiously for eruptions much larger than the ~15 Tg of $SO_2$ of the 1991 Pinatubo eruption, the biggest eruption used for calibration. EVA_H also uses the same $SO_2$ – sulfate aerosol production timescale for all eruptions, which results in too long aerosol lifetime for eruptions much smaller than Pinatubo (Vernier et al., 2024). Despite these limitations, this model captures reasonably well the magnitude and latitudinal distribution of SAOD perturbations (Aubry et al., 2020).

## 3.2 Volcanic forcing efficiency

To convert the global monthly mean SAOD at 550nm (gmSAOD) obtained from EVA_H into a global mean Effective Radiative Forcing (ERF), we use the Marshall et al. (2020) scaling:

$$ERF = \alpha(1 - e^{-\Delta gmSAOD}), \qquad\qquad (4)$$

with $\alpha$ = -20.7 ± 0.2 W.m$^{-2}$, and $\Delta gmSAOD$ corresponding only to the volcanic contribution of the SAOD. Although the

relationship gmSAOD – ERF varies with the time after eruptions, eruption latitude, and season (Marshall et al., 2020), we use a single relationship as we simulate a sequence of eruptions whose forcing commonly overlap in time. Equation 4 was calibrated using UM-UKCA interactive stratospheric aerosol model simulations covering a large range of $SO_2$ injection mass, latitude and altitude (Marshall et al., 2020). For the same gmSAOD, Eq. (4) leads to a 20-25% smaller forcing than the one estimated in IPCC AR5 (Myhre et al., 2014) (Marshall et al., 2020), but compares well to the linear relationship used in

AR6, with a coefficient of -20 ± 5 W.m$^{-2}$ (Forster et al., 2021). Whereas previous studies typically used a linear gmSAOD-ERF scaling (e.g., Schmidt et al., 2018), Marshall et al. (2020) show that a non-linear relationship better capture the simulated forcing and is more consistent physically.

### 3.3 Climate model: FaIR

To simulate the GMST of the past 9000 years, we use the FaIR reduced-complexity climate model (v2.1.4) which

reproduces well the observed temperature of the last 150 years (Smith et al., 2018; Leach et al., 2021). FaIR is a three-box energy balance model, whose parameters are the heat capacity, the heat exchange coefficient for each box, the climate feedback parameter, and the forcing efficacy, along with parameters describing responses of different forcings to their emissions. In FaIR, we consider a forcing efficacy of 1 ± 0.15 but we note that Günther et al. (2022) suggested an efficacy of 0.7 at decadal timescales. We force FaIR using GHGs, ice sheets, land-use, orbital, solar, and volcanic forcing (Sect. 2.1 and

2.2) and, from 1750 CE onwards, anthropogenic aerosol and ozone forcings. FaIR is calibrated following three steps (Smith et al., 2024): i) calibration of the model parameters against a range of CMIP6 experiments (e.g., 4×CO$_2$ experiments) for up to 49 CMIP6 models depending on component; ii) sampling of 45 parameters within the distributions of values informed by the calibrations determined in (i), generating a 1.6 million-member prior ensemble; iii) constraining the different calibrations obtained in (ii) deriving a final posterior of 1000 ensemble members by comparing them to historical observations for 1850 –

2022 and to key climate metrics from IPCC AR6. Here we use calibration version 1.4.2, which is based on CMIP6 historical



emissions. As FaIR is calibrated for an equilibrium climate corresponding to 1750 forcings and our simulations start in 6755 BCE, we spin-up the model for 1000 years with fixed forcing and concentrations corresponding to their values in 6755 BCE.

**Conversion of global annual mean temperature in a NH MJJA temperature**

The King et al. reconstruction gives a timeseries of the NH extratropical summer (NH MJJA) temperature. The four other tree-ring reconstructions and simulations from Marshall et al. (2024) and Lücke et al. (2023) estimate 40-75°N land only MJJA temperatures. To compare our model outputs to these reconstructions and simulations, we scale the FaIR simulated GMST into a NH MJJA land only temperature. We obtain a scaling factor of 1.32 from a linear regression of these two temperature metrics from the 1850 – 2017 Cowtan and Way (2014) dataset ($R^2 = 0.75$). This approach is simplistic and biased by the anthropogenic emissions-induced cooling in the second half of the 20[th] over the NH. Performing the regression only on the period 1850-1950 leads to a decrease of the slope coefficient by 15% (21% for a regression over 1850-1900) (Fig. S1).

## 3.4 Resampling strategy for volcanic injection and model parameters

To propagate volcanic injection uncertainties and modelling uncertainties, we produce an ensemble of 1000 simulations where injection and model parameters are resampled simultaneously using a Monte Carlo approach (Figure 2). Although we do not resample uncertainties on forcings other than volcanic forcing, we produced one 1000-ensemble for each of the land-use datasets considered (Sect. 2.2).

For volcanic injection parameters, we resample the mass, latitude and altitude of injection within their uncertainties, assuming Gaussian distributions. Uncertainties on the $SO_2$ mass are found in the VSSI datasets. For unidentified eruptions, we choose an uncertainty of 20° on the estimated latitude (Sect. 2.1). Regarding the injection height, if it is known an identified eruption, we use a 1-σ uncertainty of 25% of this height. For unidentified eruptions, we use a 1-σ uncertainty of 33% of our estimated height (see Sect. 2.1). For EVA_H, we sample one of the 100 optimal parameter sets provided by Aubry et al. (2020). For the SAOD-ERF scaling factor (Sect. 3.2), α is resample within its uncertainty for each run. In a run, the same factor is used for every eruption. For FaIR, we sample one of the 1000 parameter set provided by Smith et al. (2024) (Sect. 3.3). Overall, our design thus propagate uncertainty on: i) volcanic injection parameter; ii) volcanic aerosol forcing modelling; iii) climate sensitivity and uncertainty; and iv) internal climate variability.

## 4 Results

Figure 3.a-b shows our simulated gmSAOD and ERF, characterized by large changes following volcanic eruptions. The 6755 BCE – 1900 CE mean gmSAOD and ERF are $0.011 \pm 0.001$ and $-0.15 \pm 0.02$ W.m$^{-2}$, respectively, with maximum monthly values of $0.8 \pm 0.2$ and $-11 \pm 2$ W.m$^{-2}$ reached after the 5229 BCE (7179 BP) eruption (all uncertainties expressed as 1-σ) (Table 1). Although governed by individual eruptions, volcanic forcings exhibit clear long-term variability (Table 1). Millennial forcing averages vary by a factor 2, with a minimum for the third millennium BCE (0.008 mean gmSAOD, -0.08



W.m$^{-2}$ mean ERF) and a maximum for the 6$^{th}$ millennium BCE (0.013 mean gmSAOD, -0.18 W.m$^{-2}$ mean ERF), which contains the two largest eruptions since 6000 BCE (8000 BP).

**Figure 3: a. Global mean SAOD between 6755 BCE and 1900 CE; b. Corresponding global mean effective radiative forcing; c. SAOD for a selected 50-year period (1780–1830 CE), with the global mean shown in d.**





Volcanic injections with known eruption match and thus better constrained latitude and altitude have smaller forcing
uncertainties, e.g. the relative uncertainty on the peak gmSAOD following the Tambora 1815 CE eruption is 18%, whereas it
is 27% for the unidentified eruption in 1809 (Fig. 3.d). Figure 3.c illustrates that the SAOD perturbation is largely confined
to a single hemisphere for extra-tropical eruptions (e.g. Laki 1783–1784 CE), but not for tropical eruptions (e.g. Tambora
1815 CE). Furthermore, with EVA_H accounting for the effect of injection latitude and altitude, the magnitude of the
gmSAOD perturbation is smaller for lower altitude and higher latitude injections. For example, whereas Laki 1783–1784 CE
(42 Tg $SO_2$, 64°N, 9–18 km) injects 26% less $SO_2$ than Tambora 1815 (56 Tg $SO_2$, 8°N, 30 km), the 3-year mean gmSAOD
perturbation after the eruption is 70% smaller (Fig. 3.d).

**Table 1: Average global mean SAOD, volcanic forcing and volcanic cooling over different periods.**

| Time | Global mean SAOD | Volcanic forcing (W.m$^{-2}$) | Volcanic cooling |
|---|---|---|---|
| Full time series | 0.011 ± 0.001 | -0.15 ± 0.02 | -0.12 ± 0.04 |
| 6755-6000 BCE | 0.016 ± 0.002 | -0.22 ± 0.03 | -0.16 ± 0.05 |
| 6000-5000 BCE | 0.013 ± 0.001 | -0.18 ± 0.02 | -0.15 ± 0.05 |
| 5000-4000 BCE | 0.013 ± 0.001 | -0.17 ± 0.02 | -0.14 ± 0.05 |
| 4000-3000 BCE | 0.011 ± 0.001 | -0.14 ± 0.02 | -0.11 ± 0.04 |
| 3000-2000 BCE | 0.008 ± 0.001 | -0.08 ± 0.01 | -0.07 ± 0.03 |
| 2000-1000 BCE | 0.011 ± 0.001 | -0.14 ± 0.02 | -0.11 ± 0.03 |
| 1000 BCE – 0 CE | 0.010 ± 0.001 | -0.13 ± 0.02 | -0.10 ± 0.03 |
| 0-1000 CE | 0.009 ± 0.001 | -0.11 ± 0.01 | -0.09 ± 0.03 |
| 1000-1900 CE | 0.012 ± 0.001 | -0.17 ± 0.02 | -0.13 ± 0.04 |

## 4.2 Surface temperature

Our simulated GMST has an overall warming trend with superimposed annual-millennial scale variability (Fig. 4.a). Figure
4.b shows the contribution of different forcings to the simulated temperature, and the following subsections specifically
discuss the simulated millennial, centennial and decadal scale variability and how it compares to available reconstructions.
The volcanic forcing results in a cooling with a strong interannual variability. We have a mean annual cooling of -0.12 ±
0.04 K over the period 6755 BCE – 1900 CE. The maximum cooling is -2.1 ± 0.5 K following the 5229 BCE eruption.





**Figure 4: a. GMST from the full simulation ensemble (Sect. 3), expressed as an anomaly with respect to 1850-1900 CE; b. Simulated GMST with only one set of forcing varying, and all other ones set to their 6755 BCE value. The EVA_H and FaIR models were run using a single set of model parameters. Best estimates were used for volcanic injection parameters.**



### 4.2.1 Millennial-scale variability

Our simulations, the TraCE 21k simulations and LGMR reconstruction show a warming throughout the past 8500 years, with a trend of $0.08 \pm 0.03$ K.kyr$^{-1}$, +0.09 K.kyr$^{-1}$, and +0.06 $\pm$ 0.04 K.kyr$^{-1}$ respectively (Fig. 5). In contrast, the Holocene data assimilation only exhibits a warming trend over 6600 BCE – 4200 BCE (+0.15 $\pm$ 0.02 K.kyr$^{-1}$), which is then followed by a small cooling trend (-0.02 $\pm$ 0.01 K.kyr$^{-1}$). The multimillennial trend in our simulations is linked to the response to GHGs (around +0.4 K between 6755 BCE and preindustrial temperature), ice sheets (+0.4 K), orbital (+0.1 K) and land-use (-0.1 K) forcings. At a millennial scale, the mean volcanic cooling is -0.12 K, the same order of magnitude as the GHG-induced warming. Volcanic cooling averaged over a millennium can vary by a factor 2, with a maximum cooling during the 6$^{th}$ millennium BCE (-0.15 K) and a minimum cooling during the 3$^{rd}$ millennium BCE (-0.07 K) (Table 1).

Despite an excellent agreement with LGMR for the 6600 BCE – 1800 CE trend, millennial-scale variability sometimes differ between our simulations and reconstructions. Notably, we simulate a warming between 4500 and 2000 BCE (+0.16 $\pm$ 0.05 K.kyr$^{-1}$), whereas reconstructions assimilating proxy records show a stable GMST or slight cooling (-0.04 $\pm$ 0.013 K.kyr$^{-1}$ and -0.03 $\pm$ 0.03 K.kyr$^{-1}$ for LGMR and Holocene data assimilation respectively).

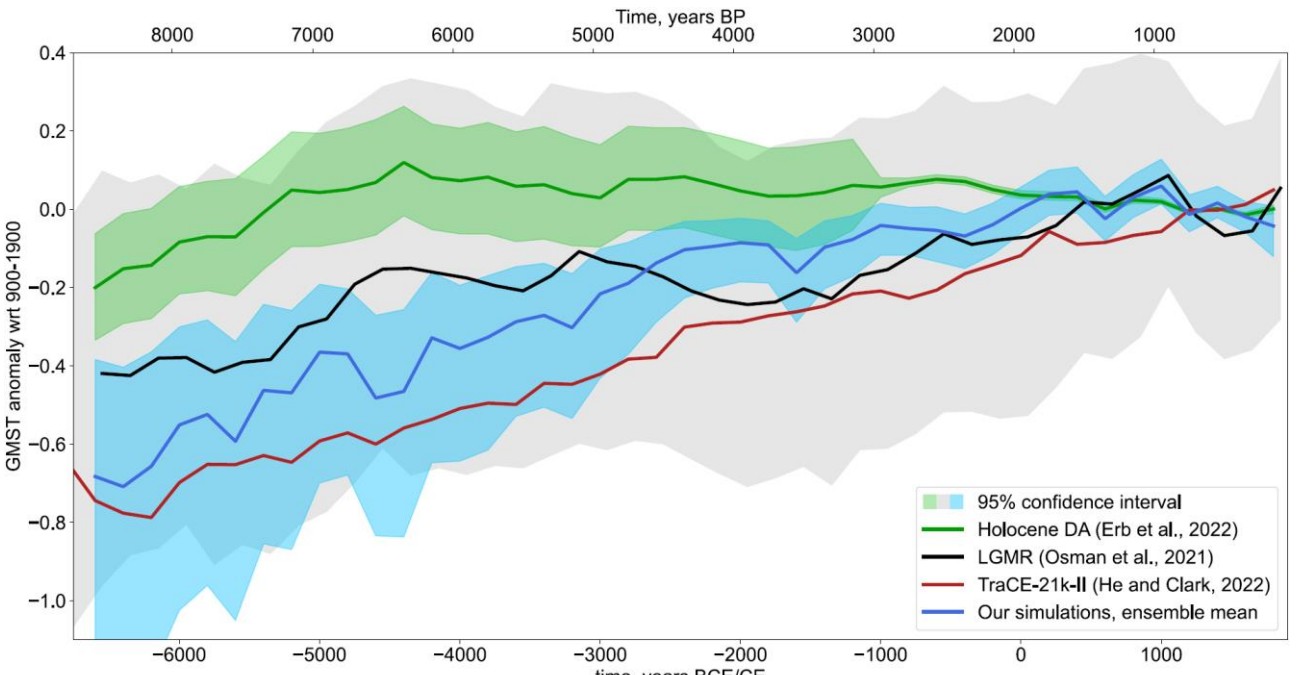

**Figure 5: Comparison of our simulated 200-year-smoothed temperatures to the LGMR (Osman et al., 2021) and TraCE-21k-II (He and Clark, 2022), expressed as an anomaly with respect to the last millennium mean temperature.**





### 4.2.2 Centennial-scale variability

Comparing our simulations to proxy-based reconstructions for the last two millennia shows some discrepancies on a centennial scale. Notably, our simulations show a weaker transition between the warm Medieval Climate Anomaly (MCA) and the cold Little Ice Age (LIA) (Table S2). The IPCC's fifth assessment report define the MCA as 950 – 1250 CE, and the LIA as 1450 – 1850 CE (Masson-Delmotte et al., 2013). Owing to the LGMR 200-year time resolution, we instead use 950-1150 CE and 1350-1750 CE.

The GMST difference between the MCA and LIA is 50% stronger in PAGES2k (0.16 ± 0.12 K, Fig. 6.a) and LGMR (0.15 ± 0.19 K, Fig. 5, noting that the duration of MCA and LIA are comparable to the time resolution of LGMR) compared to our simulations (0.10 ± 0.04 K, Fig. 6.a). The Holocene data assimilation exhibits no MCA-LIA transition (0.01 ± 0.01 K, Fig. 5). These results are robust to the choice of date to define MCA and LIA (Table S2). Compared to GMST reconstructions, the MCA-LIA transition is enhanced in some NH MJJA tree-ring based temperature reconstructions (ranging between 0.15 and 0.32 K for NTREND 2015, Guillet 2017, Büntgen 2021 and King 2021, Fig. 6.b, Table S2), but not in our simplistic simulated NH MJJA temperature estimates (0.13 ± 0.06 K, Fig. 6.b). Lücke et al.'s simulations exhibit a cooling similar to our simulations (0.12 ± 0.08 K and 0.11 ± 0.08 K, for GMST and NH MJJA land temperature respectively, Fig. 6 and Table S2).





Figure 6: a. GMST for the past 2 kyr from our simulations compared to the PAGES 2k reconstruction and the simulations from Lücke et al. (2023); b. NH MJJA temperature for the past millennium from our simulations compared to tree-ring based reconstructions (mean of NTREND 2015, Schneider 2015, Guillet 2017 and Büntgen 2021 (solid line), and King et al., 2021 (dotted line)) and the simulations from Lücke et al. (2023).





### 4.2.3 Decadal-scale variability

For individual eruptions, we simulate a cooling that lasts for 5 to 15 years for the largest eruptions. The maximum volcano-induced GMST cooling simulated is -2.0 ± 0.5 K following the 5229 BCE eruption. In terms of GMST response, we obtain an excellent agreement between our simulations and simulations performed with ESMs from Marshall et al. (2024) (Fig. 7.a
and b). When compared to reconstructions, no model-reconstruction disagreement cannot be explained by uncertainties, even for large eruptions (e.g., Samalas 1257 eruption, Fig. 7.c). Our simulated volcanic cooling is significantly larger than PAGES2K, but this dataset has previously been demonstrated to not capture abrupt temperature changes owing to the proxy assimilated and assimilation methods (e.g., Anchukaitis and Smerdon, 2022).

The superposed epoch analysis on 14 large eruptions (> 20 Tg of $SO_2$, including clustered eruptions, that occurred between
1250 and 1850 CE) shows an excellent agreement on the NH MJJA peak cooling magnitude (-0.59 ± 0.15 K in our simulations, between -0.30 and -0.56 K in reconstructions, Fig. 7.d, and a mean cooling of -0.35 ± 0.08 K and between -0.18 and -0.33 K respectively over the first five years after eruption). Lücke et al. and Marshall et al. obtain a NH land-only MJJA peak cooling 56% and 76% stronger than our simulations (-0.92 and 1.04 K respectively). The year of recovery, defined as the first post-eruption year with a cooling less than 5% of the peak cooling, is in excellent agreement in our simulations (12
± 3 years), tree-ring based reconstructions (15 years for King 2021 and NTREND 2015, 13 and 18 years for Guillet 2017 and Büntgen 2021 respectively), and aerosol properties driven simulations from Marshall et al. (2024) (12 ± 1 years). $SO_2$ emission driven simulations from Marshall et al. (2024), simulations from Lücke et al. and the Schneider 2015 tree ring reconstruction recover earlier (10 ± 2, 8 ± 1 and 8 years respectively). Defining the post-eruption warming trend as the linear trend between the year of peak cooling and year of recovery, we obtain a good agreement between reconstruction (between
0.022 and 0.036 K.yr$^{-1}$), aerosol properties driven simulations from Marshall et al. (2024) (0.035 K.yr$^{-1}$) and our simulations (0.036 ± 0.07 K.yr$^{-1}$), with Lücke et al.'s and $SO_2$ emission driven simulations from Marshall et al. (2024) exhibiting a stronger trend (0.049 and 0.048 K.yr$^{-1}$ respectively). Note that, because of clustered eruptions, this does not reflect the mean response to a single eruption (see Fig. S2 for a superposed epoch analysis with no cluster).





**Figure 7: a. GMST response to Samalas 1257 eruption from our simulations compared to simulations from Marshall et al. (2024) and Lücke et al. (2023); b. superposed epoch analysis of the GMST response to 14 eruptions injecting more than 20 Tg of SO2 that occurred between 1250 and 1850; c. NH land only MJJA temperature response to Samalas compared to tree-ring-based reconstructions (NTREND 2015, Schneider 2015, Guillet 2017, Büntgen 2021, King et al., 2021) and simulations from Lücke et al. (2023) and Marshall et al. (2024); d. Superposed epoch analysis of the NH land only MJJA temperature response. *Note that King et al.'s reconstruction corresponds to land and sea mean temperature, whereas other reconstructions are land only.**


## 5 Discussion

### 5.1 Sensitivity to volcanic aerosols and the climate model

The simulated annual GMST response to large volcanic eruptions is on average in good agreement between Lücke's simulations and ours (Fig. 7.b). This result is surprising because our use of EVA_H instead of EVA for aerosol modelling results in lower global mean SAOD estimates on average by 25% (Fig. S3.b). This is primarily caused by different






calibration datasets (Aubry et al., 2020). EVA_H predicts a peak global mean SAOD that is 25% smaller than EVA for Pinatubo-like (tropical, high-altitude) volcanic injections (e.g. Fig. S3.a, Tambora 1815). This smaller forcing in our simulations results in minor differences in terms of average climate response, likely as a consequence of compensating differences in climate models. With FaIR, our study resamples parameters to cover the full spectrum of CMIP6 climate

model behaviour, whereas in Lücke et al. their reduced complexity climate model is only tuned against HadCM3 simulations. Our simulation ensemble with FaIR will capture the HadCM behaviour thanks to our resampling strategy (including FaIR parameters, Sect. 3.3), but our predicted mean climate response will be a lot closer to the CMIP6 multimodel mean. The smaller discrepancy in the climate response than in the SAOD estimate can also be linked to the sampling of the date of eruption in Lücke et al. (2023) for unidentified eruptions. Applying the same eruption date sampling

method in our workflow to an unknown eruption (1809 eruption) results in peak gmSAOD and cooling 30% and 20% smaller respectively than our simulations without date uncertainty sampling (Fig. S4).

The main exception to the overall agreement in post-eruption GMST decrease between our study and Lücke et al. is for high-latitude Icelandic eruptions (e.g. Eldgja 939 and Laki 1783, Fig. S7), with differences by up to a factor 2.3. This is likely primarily caused by the fact that EVA_H accounts for the impact of both injection height and latitude on the magnitude and

e-folding time of volcanic SAOD perturbations (Aubry et al., 2020), which can result in differences in SAOD by a factor of 2 for such eruptions (e.g. Fig. S3.a, Laki 1783–1784). Last, we note that our method to compare reconstruction and simulated NH MJJA temperatures relies on a simplistic scaling of the annual GMST simulated using the global mean SAOD. Lücke et al. use a more sophisticated approach, for land-only NH MJJA temperatures, with their reduced-complexity model calibrated to emulate the land-only NH MJJA temperature simulated by HadCM3 based on the NH mean SAOD. This

method produces land-only NH MJJA temperature responses stronger by a factor 1.6 (Fig. 7.b) to 3 (Fig. S6, for NH eruptions) than the land only NH MJJA temperature directly scaled from annual GMST. Such calibration for FaIR using all CMIP6 models was beyond the scope of our study. We note that our simpler method shows good performances when comparing both our NH MJJA temperature responses to tree ring-based reconstructions (Sect. 4.2.3, Fig. 7.c and d).

## 5.2 Medieval Climate Anomaly – Little Ice Age transition

Our simulations underestimate the magnitude of the MCA-LIA transition compared to proxy-based reconstructions (Sect. 4.2.2, Table S2). For example, in terms of NH MJJA temperature, for our default HYDE 3.3 land-use forcing and for our default MCA and LIA definitions of 950–1250 and 1450–1850, reconstructions have a transition ranging from 0.15 to 0.32 K whereas we simulate a weaker transition at $0.13 \pm 0.06$ K. The magnitude of the transition is sensitive to the choice of exact period to define the MCA and LIA (Table S2), and we note that the Schneider 2015 reconstruction does not show a

transition on the considered periods, but the fact that our simulation underestimate it is robust to it. However, our simulations are in better agreement with global-scale data assimilation methods and in terms of global mean temperature, for which our simulated transition is only 35% weaker than LGMR and PAGES2k. This is expected since we simulate global mean temperature and the MCA-LIA transition was not a global phenomenon (Mann et al., 2009; Wanner et al., 2022) or not



synchronous on different continents (Masson-Delmotte et al., 2013). Discrepancies remaining in terms of global mean
temperature might be due to the fact that even global reconstructions mostly assimilate NH land summer temperatures proxy. Although the simplicity of our modelling approach might be another cause, the MCA-LIA transition simulated by CMIP5 GCMs in terms of NH temperature is also twice as small as proxy-based reconstructions (Masson-Delmotte et al., 2013). Last, model-reconstruction discrepancy could simply be explained by forcing biases. In particular, the choice of the land-use forcing dataset (Sect. 2.2) strongly modulates the magnitude of the simulated transition (Table S2), with weaker transition
using HYDE 3.2 (Klein Goldewijk et al., 2017) and stronger transition using KK10 (Kaplan et al., 2011), which has a steeper land-use forcing trend in particular outside Europe (ArchaeoGLOBE Project, 2019). When using KK10, our simulated MCA-LIA transition agrees with any considered reconstruction within uncertainty. The only exception is the Holocene Data assimilation in which no MCA-LIA is visible which can be attributed to the low resolution and age synchronization of the underlying proxies (Kaufman et al., 2020; Esper et al., 2024) and the absence of volcanic and solar forcing in the model prior
which are considered the main drivers of the LIA (Mann et al., 2009; Owens et al., 2017; Wanner et al., 2022).

### 5.3 Millennial climate trends

We obtain a good agreement between proxy-based reconstructions and simulations in terms of the multimillennial trend, but not for the period 4500 – 1000 BCE. Regardless of the land-use forcing used (Fig. S5), our simulations exhibit a relatively steady warming, whereas reconstructed temperatures are relatively stable or slightly cooling in both LGMR and the
Holocene DA (Fig. 5). This reconstruction-simulation discrepancy, known as the Holocene temperature conundrum, is not unique to our simulations and has been extensively discussed in the literature (e.g., Liu et al., 2014; Hopcroft et al., 2023). Kaufman and Broadman (2023) suggest that an increasing volcanic forcing between 4000 BCE and the last millennium could explain the cooling trend observed in proxy reconstructions. Using the same base volcanic emission dataset (Toohey and Sigl, 2017), our simulated temperature does not support this hypothesis (Table 1). This is consistent with the fact that
millennial-scale mean volcanic forcing was relatively constant between 4000 BCE – 1000 CE (around 0.12 W.m$^{-2}$) followed by a stronger forcing only during the last millennium (0.17 W.m$^{-2}$) (Table 1). Another potential missing factor is the greening of Sahara (between 7000 and 4000 BCE, Hoelzmann et al., 1998). Our reduced-complexity model cannot reproduce this transition nor associated climate feedback, and the land-use forcing datasets we use only account for anthropogenic land-use change. Thompson et al. (2022) show that including a vegetation cover over the Sahara between
7000 and 4000 BCE could explain the warming until 4000 BCE and following cooling, comparable to the trends observed in the Holocene data assimilation (+0.15 ± 0.02 K.kyr$^{-1}$ for 6600 – 4200 BCE and -0.02 ± 0.01 K.kyr$^{-1}$ for 4000 BCE – present).



**5.4 Using reduced-complexity climate models to estimate the climate effects of volcanic eruptions**

This paper demonstrates the skills of our reduced-complexity modelling framework to investigate past climate change at annual-multimillennial scale, and in particular the climate impacts of volcanic eruptions. We find:

- An excellent agreement on annual to decadal timescales for the mean temperature response to volcanic eruptions (superposed epoch analysis, Fig. 7.b, Sect. 4.2.3);
- For individual eruptions, there is no discrepancies that cannot be explained by uncertainties (Fig. 7.a, Fig. S6, Fig.

S7, Sect. 4.2.3);
- A good agreement on the multimillennial trend (Fig. 5, Sect. 4.2.1).

Although the aerosol (EVA_H) and climate (FaIR) models are empirical, these successes are worth noting since: i) large eruptions, in particular the one used in the superposed epoch analysis (18-118 Tg $SO_2$), far exceed the 15 Tg $SO_2$ of the Pinatubo 1991 eruption, the largest eruption used for EVA_H calibration; ii) FaIR was mostly calibrated using simulations

with zeroed or constant volcanic forcing (Sect. 3.3) such as 4×$CO_2$ simulations and validated against the post-1850 era, and it was not calibrated using any paleoclimate simulation. Our modelling framework presents the advantage of a negligible computing cost, allowing to simulate long climate time periods and/or large simulation ensembles to propagate uncertainties related to input forcings and models. Our study thus supports the continued use of models like EVA_H and FaIR, in parallel to ESMs and interactive stratospheric aerosol models.

One of the main limitations of our modelling framework is that it is currently restricted to simulation of GMST, whereas comparison to proxy reconstructions often require regional and seasonal signal, and societal impacts require assessment of regional climate response and other variables like precipitation. For example, to compare our simulations to tree-ring-based reconstructions, we used simplistic empirical scaling between GMST and NH MJJA temperatures based on the historical temperature record (Sect. 3.3, Fig. S1). To make progress and simulate the response of diverse climate metrics at regional,

seasonal scales whilst maintaining a low computing cost, future work could use methods developed for spatial emulation of ESMs. These methods include pattern scaling (e.g., Mathison et al., 2024) and machine learning methods (e.g., Watson-Parris et al., 2022), which to date have mostly been applied to investigate climate response to anthropogenic forcings, but not volcanic forcing.

**6 Conclusions**

By combining reduced-complexity volcanic aerosol (EVA_H) and climate (FaIR) models, we produced a 1000-simulation ensemble of the 6755 BCE – 1900 CE (8705 – 50 BP) temperatures, resampling uncertainties on the volcanic emissions, aerosol and climate models, and internal climate variability. We also account for the temperature variation induced by GHGs, solar, orbital, land-use and ice sheets related forcings. We compare these simulations to proxy-based reconstructions both for global annual mean and NH summer temperature. We find a 6755 BCE – 1900 CE mean volcanic forcing of -0.15

W.m$^{-2}$ and volcanic cooling of -0.12 K, and quantify millennial-scale variations in volcanic cooling and forcing (1000-year



average ranging between -0.08 and -0.22 W.m$^{-2}$, and -0.07 and -0.16 K respectively). We obtain an excellent agreement between tree ring-based reconstructions and our simulations for the mean surface temperature response to the largest eruptions of the last millennium. For individual eruptions, we show that discrepancies between reconstructions and simulations can be explained by the uncertainties on volcanic emissions, aerosol and model parameters, and internal climate

variability. At longer timescales, we obtain a good agreement between proxy-based reconstructions and simulations in terms of multimillennial trend. We note important discrepancies between our simulations and reconstructions in terms of centennial-millennial scale variability, which also affect more complex GCMs and ESMs. Overall, this study supports the use of reduced-complexity aerosol and climate model to estimate the climate response to volcanic eruptions. This approach perfectly complements studies with ESMs with or without interactive stratospheric aerosol scheme, as its low computational

cost enables an extensive propagation of uncertainties related to both model inputs and model themselves, as well as the simulation of long timeseries.

**Code availability**

EVA_H is available from https://github.com/thomasaubry/EVA_H.
FaIR v.2.1.4 is available from https://github.com/OMS-NetZero/FAIR/tree/v2.1.4.

**Data availability**

gmSAOD, ERF and volcanic simulations ensembles are available from https://doi.org/10.5281/zenodo.14170014. Volcanic emissions reconstructions are available from https://doi.pangaea.de/10.1594/PANGAEA.928646 (HolVol v1.0) and https://doi.org/10.26050/WDCC/eVolv2k_v3 (eVolv2k v3). Orbital, solar, ice sheets and land use forcing reconstructions
were provided by P.O. Hopcroft and corresponding reconstructions are available from https://doi.org/10.17617/1.5U (solar irradiance) https://geo.public.data.uu.nl/vault-hyde/HYDE%203.3[1710493486] (HYDE 3.3), https://doi.org/10.17026/dans-25g-gez3 (HYDE 3.2), and https://doi.pangaea.de/10.1594/PANGAEA.871369 (KK10).

Most of the proxy reconstructions are publicly available from the NOAA National Centers for Environmental Information (NCEI), under the World Data Service (WDS) for Paleoclimatology at https://www.ncei.noaa.gov/access/paleo-search/
(Schneider et al., 2015, https://doi.org/10.25921/6mdt-5246; Wilson et al., 2016, https://doi.org/10.25921/kztr-jd59; Guillet et al., 2017, https://doi.org/10.25921/42gh-z167; PAGES 2k Consortium, 2019, https://doi.org/10.25921/tkxp-vn12; Büntgen et al., 2021, https://doi.org/10.25921/9986-r929; King et al., 2021, https://doi.org/10.25921/vey7-kx38; Osman et al., 2021, https://doi.org/10.25921/njxd-hg08). The reconstruction from Erb et al. (2022) is available on Zenodo at https://doi.org/10.5281/zenodo.6426332.



Simulations from Marshall et al. (2024) are available from the supplementary materials of https://doi.org/10.5194/egusphere-2024-1322. Simulations from Lücke et al. (2023) are available from https://doi.org/10.7488/ds/3834. TraCE-21K-II model data are available from https://trace-21k.nelson.wisc.edu.

**Author contribution**

Conceptualisation: TJA, MV
Data Curation: MV, CS, POH
Formal analysis: MV
Funding acquisition: TJA, CS
Investigation: MV
Methodology: MV, TJA, CS, POH, MS
Supervision: TJA, CS
Visualisation: MV
Writing – original draft preparation: MV, TJA
Writing – review and editing: All authors

**Competing interests**

The authors declare that they have no conflicts of interest.

**Acknowledgements**

We sincerely thank professor Kees Klein Goldewijk for facilitating early access to version 3.3 of the HYDE dataset.

**Financial support**

MV acknowledges funding from the University of Exeter through PhD scholarship and research funding from Exeter's
Department of Earth and Environmental Sciences (DEES), the Met Office through a CASE studentship, GW4 generator grant (no. GW4-GF4-018) and IAVCEI Commission of tephrochronology. TJA acknowledges support from the University of Arizona through a Haury Fellowship, a travel award from the Canada-UK foundation, and the Exeter global travel fund, DEES, Camborne School of Mines International Travel Bursary. MS acknowledges funding from the European Research Council under the European Union's Horizon 2020 research and innovation program (grant agreement no. 820047).



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
