# Peer review of "Using reduced-complexity volcanic aerosol and climate models to produce large ensemble simulations of Holocene temperature"

_EGUsphere, 2024_

## Author Comment (AC1)

The authors use a reduced-complexity climate model to obtain an ensemble of simulations of the last 9000 years with different volcanic forcing for the purpose of quantifying volcanic forcing uncertainty, and to put them into the context of proxy reconstructions. This is a nice and well written paper, but I have some concerns about their conclusions. I hope the authors will accept my constructive criticism which is meant to increase the novelty of the manuscript and to strengthen their results.

Major remarks:

(1) My main point of concern is the fact that the NH climate has been simply scaled to match the observed record for the 20th century. While the authors have noted in the discussion that this approach is simplistic, two of their three major conclusions rely heavily on their NH temperature simulation and thus on a simple scaling approach.

The problems I see are:

- Scaling suppresses the amplitude of the record and thus the response to volcanic eruptions.

- However we do not know if the same scaling applies during periods of volcanic activity.

- In fact volcanic activity is almost negligible within the 167 yr record considered for the linear regression.

- Thus the scaling approach is not validated for representing volcanic years, however this is the main purpose of the paper/

In short, we may see a better fit between the here presented simulations and proxy reconstructions due to the suppressed amplitude, but this can likely be an artefact of the scaling approach. Their results therefore need to be strongly caveated, which significantly reduces the novelty of this manuscript. I would suggest to either obtain a better way of quantifying NH climate, or to put a larger emphasis on their global results.

In order to provide a better context for their results, they could also repeat their model simulations for volcanic forcing using EVA instead of EVA_H and compare the results. This would provide a better context for comparisons with Luecke et al. 2023 and with reconstructions. If they can thus prove that the reason for the better performance results from the use of EVA_H compared to EVA, and not from their scaling, this would justify their conclusions. If the different results are a result of the superiority of the FAIR model over the model used by Luecke et al., the authors need to find a better way of estimating NH climate.

We would like to thank the reviewer for highlighting further potential caveats of this approach. To test the reviewer hypothesis and strengthen the robustness of our results, we calculated scaling factors based on simulations from six ESMs from Marshall et al. (2025). For each of the six models included in their study (UKESM1, CESM2(WACCM6ma), MRI-ESM2, MPI-ESM1-2-LR, MIROC-ES2L and IPSL-CM6A-LR), we define as "volcanic years" the 5 years following an eruption larger than Pinatubo (>7.5 Tg of S injected in the stratosphere) resulting in 79 volcanic years. The scaling factors from the linear regression between GMST and NH MJJA land only temperature, for each model, are presented in table 1. For comparison, the scaling factor we obtained from the observations from Cowtan and Way (2014) for 1850-present day (not for the 20$^{th}$ century only) was equal to 1.321.

Table 1: regression slopes (C NH MJJA land only/C annual global) for volcanic and non-volcanic years for different models from Marshall et al., 2025

| Models | Slope for volcanic years | Slope for non-volcanic years |
|---|---|---|
| UKESM1 | 1.654 | 1.237 |
| CESM2(WACCM6ma) | 1.544 | 1.418 |
| MRI-ESM2 | 1.190 | 1.046 |
| MPI-ESM1-2-LR | 1.397 | 0.999 |
| MIROC-ES2L | 1.722 | 1.137 |
| IPSL-CM6A-LR | 1.244 | 1.524 |
| Mean | 1.459 | 1.227 |

Figure 1 presents Superposed Epoch Analyses using the different scaling factors from table 1 to convert our simulated GMST into a NH MJJA land only temperature. We observe that applying a scaling characterizing volcanic years based on ESM simulations does not significantly alter our results. Using the largest scaling for volcanic years (derived from MIROC-ES2L data) leads to a peak cooling in the SEA 30% stronger than when using the scaling derived from observations, but still overlap with the uncertainty of tree ring reconstructions (fig. 1h). Overall, this analysis further highlights the simplicity of the scaling used and uncertainties on this scaling, but also suggest that the relatively good agreement obtained is not caused by a bias in the observational dataset used to derive the scaling. We included Fig 1 and Table 1 above as SI material, discuss them explicitly in the manuscript (in particular Sect. 3.3 and 5.1) to further highlight uncertainties on our simple scaling approach, and ensured that key results are formulated with sufficient caution.

[Figure]

Figure 1: SEA for different scaling factors used to convert our simulated GMST into a NH MJJA land only temperature. **a)** our simulations are scaled using a scaling factor determined by linear regression over 1850-2017 observations from Cowtan and Way (2014); **b)** using a scaling factor corresponding to the mean slope for volcanic years over six ESM simulations for 1250-1850 from Marshall et al. (2025); **c)** using the scaling factor from Cowtan and Way, multiplied by 1.19, which is the model-mean ratio between the factor derived from volcanic years and from non-volcanic years, where we used again the Marshall et al. (2025) model simulations; **d-i)** using a

scaling factor determined by linear regression over the volcanic years for each model (summarized in table 1).

Following a suggestion from reviewer 1, we added a 1000-member ensemble of simulations for the historical period (1850-2021 CE), thus strengthening our results (Sect. 4.2.4).

(2) Another point of concern is that dating uncertainty has not been taken into account. For the purpose of comparison with proxy reconstructions, this is a very important source of uncertainty, and in particular plays a major role for presenting superposed epoch analyses. Since the SEA is one of the major results presented, I would suggest to repeat the SEA but to perturb the sampled eruption year and thus get a measure of how dating uncertainty would change the amplitude of their simulations.

Most of the eruptions considered in the SEA are historical, with Greenland ice core chronologies well anchored with historic eruptions from Iceland in 1362, 1477 1693 and 1783. Consequently, out of the 14 eruptions, 9 have no age uncertainties, 2 have been found in agreement with historical accounts of dark lunar eclipse (Guillet et al., 2023), and the remaining have uncertainties of ± 1 year. The impact of uncertainty resampling on the SEA will thus be minimal. Artificially resampling a 2-year uncertainty for all eruptions reduces the peak cooling in our simulations by 20% (tested for the 1809 eruption, Sect. 5.1, Fig. S4), as discussed in our original submission. Resampling a dating uncertainty of 2 years as in Lücke et al (2023) is not justified by the uncertainty in the ice-core datasets for the 14 eruptions of the last millennium considered in our SEA.

(3) I am also intrigued that the authors have provided a full 9000 year simulation, however concentrate largely on their discussion of last Millennium climate. Would it be possible to extend their discussion of Holocene climate? As I understand this approach of using a reduced complexity model and a large forcing ensemble is completely novel for Holocene climate. I would like to have a clearer idea what we learned from this experiment!

To enrich the discussion of Holocene results, we additionally compare our simulations over the whole Holocene with simulations on MPI-ESM performed by Van Dijk et al. (2024) (see revised figures 5 and 7). However, our discussion remains more focused on the last millennium since large-scale proxy reconstructions with an annual resolution and precision do not go much further back in time. Our key results on the Holocene period are that our simple modelling framework:

1) Captures well the multimillennial trends, which is an important test given no Holocene data was used for calibration.

2) Does not capture well apparent centennial-millennial scale variability throughout the Holocene derived from proxy networks.

We agree that this second point merits further discussion, but such discussion would require significant additional analyses that are beyond the scope of our paper.

(4) I am not sure how internal variability is accounted for. It is mentioned a few times but nowhere explained in detail (or has slipped my attention).

Internal variability is a new module introduced in v2.1 of FaIR and is based on the autocorrelation of variability around a mean state that is calibrated on abrupt-4xCO2 models from CMIP6, as described mathematically in Cummins et al. (2020) and incorporated into FaIR as described in Smith et al. (2024). A sentence has been added to describe this.

(5) Lastly, I understand the authors have used different implementations of land-use forcing. As far as I can tell no results have been shown or are being discussed anywhere. Can you please clarify what the purpose of this experiment was?

We show the effect of different land use datasets in figure S5 and Table S3. The choice of land use forcing alters longer term trends (centennial to millennial). For example, as discussed in section 5.2., the amplitude of the cooling between the Medieval Warm Period and the Little Ice Age varies by a factor 3 between the lowest land use forcing (HYDE3.2) and the highest (KK10) (Table S3 and figure S5b).

Further comments:

- 104 ff. How is the eVolv2k record different to the ensemble described in Luecke et al. 2023? What is the added benefit for creating a new ensemble from scratch? I think this needs to be more clearly brought out in the introduction. Is the novelty the length? Or the new approach?

  The main differences with the ensemble described in Luecke et al 2023 are:

  1) Foremost, the choice to use EVA_H rather than EVA. This version of the reduced-complexity aerosol model is calibrated over the full satellite era (rather than just Pinatubo) and is sensitive to latitude and height (the latitudinal SAOD distribution is sensitive to latitude in EVA, but not its global mean).
  2) Since EVA_H is sensitive to the height of injection, we also added the height as one of the parameters.
  3) Compared to the original eVolv2k database, we make the choice not to include coincidental matches before 1750 as identified eruptions, resulting in the removal of 11 matches.
  4) We also added 3 recent geochemical matches, previously not included in eVolv2k.
  5) We use a new definition of the default latitude of an unidentified eruption, based on the asymmetry of the bipolar deposit (rather than the default 45°N/0°N/45°S in eVolv2k).

  Changes 3-5 were made to be consistent with the CMIP7 volcanic forcing dataset. We acknowledge that the detailed documentation of this dataset is not available yet owing to the time pressure of CMIP forcing delivery and the

need to prioritize forcing delivery over detailed dataset documentation to date. A succinct and preliminary documentation is now available as a technical note preprint and we added the reference to the manuscript.

- 3: M has not been introduced. I understand that the reference has not been published yet. But this needs further justification or context (i.e. where do the scaling factor and exponent come from?).

  We have added the following in the manuscript: "The scaling factor and exponent were determined over a compilation of satellite era eruptions, and HolVol and eVolv2k eruptions with geochemical matches, as well as geological data to constrain plume heights of ice-core events."

- Why is timing uncertainty not included???? This is a key uncertainty when comparing to proxy data, and in particular heavily affects the SEA??

  See point 2 of major remarks.

- L 265: 'EVA_H's empirical nature' – In what sense is this empirical? Is this the same for EVA? Please clarify.

  EVA_H is considered empirical because of its calibration against satellite observations. We change this sentence into "Given the range of calibration of EVA_H, results should be interpreted cautiously for eruptions much larger than the ~15 Tg of SO2 of the 1991 Pinatubo eruption, the biggest eruption used for calibration."

- L 267-269: So caution is needed for interpreting both eruptions larger than Pinatubo and much smaller than Pinatubo. How would this bias the results?

  When we refer to eruptions "much smaller than Pinatubo", we mean at least one order of magnitude smaller. Since we are working with ice-core records of volcanic eruptions, most of the eruptions in our VSSI database are larger than 1Tg of $SO_2$. The risk of bias in the results because of smaller eruption is less important. We note that no matter the complexity, and including for global interactive stratospheric aerosol model, caution should be used when using any model to quantify the forcing of eruptions that were not well observed (e.g. Clyne et al., 2021).

- L 268-269: Please can you add a quantifying statement to this.

  We edited the manuscript as follows: "Despite these limitations, this model captures reasonably well the magnitude of global mean SAOD perturbations for 1979-2014 (RMSE = 3.8 × 10$^{-3}$, compared to a mean SAOD over the period of 0.015), as well as the latitudinal and vertical aerosol distribution compared to observations from GloSSAC (Fig. 8 and 9, and latitudinal distribution of SAOD perturbations (Aubry et al., 2020)"

- L 290: anthropogenic aerosol and ozone forcings – this is new to me as has not been mentioned anywhere before? Should be added to the section about forcing. Also please add reference.

  From 1750 (approximately the start of the Industrial Era) onwards we include the influence of anthropogenic forcings, which in addition to the greenhouse gases and land use change already considered over the Holocene period, includes aerosols and ozone. The simulations are performed by running the FaIR model with emissions of short-lived forcers (i.e. aerosol and ozone precursors), as well as greenhouse gases emissions

  We added the following paragraph in the forcing section: "From 1750 CE onwards, we include the influence of anthropogenic forcings, which in addition to the greenhouse gases and land use change already considered over the Holocene period, includes aerosols and ozone. The simulations estimating these two forcings are performed by running the FaIR model with emissions of short-lived forcers (i.e. aerosol and ozone precursors), as well as greenhouse gases emissions from the RCMIP project (Nicholls et al., 2020, version 5.1.0, Nicholls and Lewis, 2021). Before 1750 CE, these forcings are set to their 1750 CE value, i.e. we consider that there is no change in anthropogenic forcing."

- L 320: I do not understand how the design accounts for internal climate variability!

  See point 4 of major remarks.

- 370 ff. Given the extensive discussion of the MCA-LIA difference it would be worth adding a boxplot figure showing the differences between the reconstructions.

  We already have a table in SI summarising this information. Given the large amount of figures and tables, we had to make choices on what to include in SI or in the main article.

- Figure 6: I wonder what cause the deviation between your simulations and Luecke et al. Is this simply a result of the anomalies and a mismatch in the 1850-1900 period or is this the result of different forcing or model differences?

  Fig. 2 shows the same comparison with the anomalies expressed with respect to 1450-1850 average. Our simulations and Luecke et al.'s appear to match well except for the last 200 years. We suspect that this might be linked to a difference in climate sensitivity between HadCM3 and the CMIP6 range.

- Section 5.: The fact that the MCA-LIA difference is much smaller in the simulation compared to the proxy reconstructions could also be an artefact of tree-ring spectral biases, in particular RW type proxies (which also explains why

Schneider does not show this difference). Please add this to the discussion and reference Luecke et al. 2019

We added: "The only proxy-based dataset with a weaker transition than our simulations is Schneider et al. (2015), which might be explained by the fact that they do not use tree-ring width data which might have a strong memory bias leading to an overestimation of long-term anomalies (Lücke et al., 2019)."

- Section 5.4 has a large overlap with the conclusions and could be merged.

Thank you for this suggestion. We have now merged these two sections.

Minor remarks:

- L18-19: What about dating uncertainty?

Not accounted for, see point 2 of major remarks

- L 22: So no anthropogenic aerosols?

Anthropogenic aerosols only accounted for over 1750-1900 (forcing = 0 before 1750), idem for ozone forcing. Text modified as follow: "accounting for volcanic forcing, solar irradiance, orbital, ice sheet, greenhouse gases, land-use forcing, and anthropogenic aerosols and ozone forcing for the historical period (1750-1900 CE)."

- L 25: is this averaged over 9000 years?

We change this sentence into "averaging over the last 9000 years, we obtain…"

- L 32: "a relatively cool period in climate reanalyses" – Wording is unclear. Do you mean the cool period is found in climate reanalyses or is the cooling not found in reanalyses?

Change for "we also do not capture a relatively cool period between 3000 BCE and 1000 BCE visible in climate reanalyses".

- L 46: tree ring data is the only proxy with reliable annual resolution for the last Millennium.

We respectfully disagree on this. Tree-rings arguably are the best dated proxies, but many other proxies have annual resolution as well. Ice-cores for example have annual resolution and over the last millennium have no age uncertainty relative to the volcanic forcing also derived from ice cores. We therefore decided to keep the sentence general to also include other proxies.

- L 47: Cite Luecke et al. 2021 for seasonal bias

Done, thanks for the suggestion.

- L 53: Cite Luecke et al. 2023 for volcanic forcing uncertainty

  Here we are citing paper referring to the uncertainties in the emissions, rather than how modelling workflows have included these uncertainties. Lücke et al. (2023) is not a paper documenting the ice-core dataset, hence the omission. Marshall et al. (2019) is a modelling study, but they illuminate uncertainties in deposition factors used to convert deposition to mass in ice-core datasets.

- L 72: Do you mean "...we refer to 'reduced-complexity models' as idealised models"?

  Thanks for pointing that, we corrected this.

- L 126: I'm not sure if this is due to the draft setting, but the formatting here is really off (also in the following equations)

- L 127: asym is in math mode (italic) but should be text mode (upright) in eq. (1)

  Done.

- L 131: Is it worth quantifying the uncertainty for identified eruptions? I assume there is significant uncertainty associated with those estimates of plume height. It would be interesting to know how this compares with unidentified eruptions uncertainty.

  We do quantify the uncertainty over the height for identified eruptions, since different methods to estimate can lead to a wide range of estimated heights. For example, for the Samalas 1257 eruption, the estimated top height based on isopleth ranges between 38 and 59 km high (25.5 to 39.6 for the $SO_2$ injection height). To sample the full range of possible height, we apply an uncertainty of 25% of the height for identified eruptions with a known injection height (Sect 3.4). For unidentified eruptions, we estimate a height based on the mass of $SO_2$ (Sect. 2.1) and apply an uncertainty of 33% (Sect 3.4). We acknowledge that these numbers are ad-hoc, and it would be preferable to have bespoke height uncertainties for each known eruption. However, this information is rarely available in the literature.

- L 131: Is this correction about identified eruptions? This is a bit hard to follow for anyone unfamiliar with isopleth maps.

  Yes, it is. Isopleth maps is a classic way to estimate the top plume height based on the distribution and clast size of volcanic deposits. However, using the IVESPA observational eruption source parameter database, Engwell, Aubry et al. (2023) showed that this height metrics overestimates the SO2 injection height by 49% on average. To obtain the SO2 injection height, we divided the height derived from isopleth maps by 1.49. We clarified the text by changing "when the height is derived..." into "if this height..."

- L140/eq. 3: What is M? This has not been introduced.

  We introduced it on line 141: "default mass of SO2 ($M$)".

- L 141: confusing wording: Fig. 1f provides an overview but emission parameters are in SI? This is unclear. Also include reference to where in SI it's shown.

  Figure 1f shows the spatiotemporal distribution of VSSI. We provide the link to the zenodo repository containing a spreadsheet of the VSSI used to build this figure and our ensemble of simulations. The sentence now reads "Figure 1.f provides an overview of the final volcanic SO2 emission parameters we used (a detailed table of our volcanic emission parameters is available from https://doi.org/10.5281/zenodo.14170014) after application of above methods."

- L 201-209: No remark but I really like this brief discussion of limitations of the proxy records.

  Thank you for the kind comment.

- This discussion of the key differences to Luecke et al.'s study should have been discussed at least briefly in the introduction.

  We added the following in the introduction: "Compared to Lücke et al. (2023), our approach extends simulation by 8000 years, makes use of a reduced-complexity aerosol model accounting for volcanic emission latitude and altitude (EVA_H), and of a reduced-complexity climate model enabling to emulate the full range of behaviour of CMIP climate models as well as internal climate variability (FaIR)."

- L 258: 'EVA_H accounts for …'

  Corrected, thank you

- L 274: put \alpha in math mode

  Done

- (4) put -\Delta gmSAOD into text mode

  Done

- L 275: It would be better to write this in text, i.e. 'the relationship between gmSAOD and ERF' otherwise it could be interpreted as gmSAOD minus ERF.

  Thanks for helping us avoid a confusing wording, corrected.

- L 276 Eq. (4)

  The convention in CP is that at the beginning of a sentence, the unabbreviated word "Equation" should be used but thanks for reminding us of the use of parenthesis.

- L 294-295: does this ensemble represent different values for climate sensitivity? Please clarify.

  Yes, it does. This ensemble is designed to fit the IPCC AR6 WG1 distribution of a range of climate metrics, including Equilibrium Climate Sensitivity (see Table 6 in Smith et al., 2024). We clarify the text as follow: "iii) constraining the different calibrations obtained in (ii) deriving a final posterior of 1000 ensemble members by comparing them to historical observations for 1850 – 2022 and constraining key climate metrics (e.g. equilibrium climate sensitivity) to the range of values from IPCC AR6."

- L 327-328: Okay but uncertainty is also very large for these eruptions. So within the uncertainty range (assuming the gmSAOD uncertainty is symmetrical around the mean), more recent eruptions could in fact exceed them.

  Indeed, within the uncertainty range, it is possible that more recent eruptions could exceed them. For comparison we add the gmSAOD/ERF/temperature anomaly values for the 1257 Samalas eruption.  (for comparison, we obtain a gmSAOD of 0.41 ± 0.08 and an ERF of -6.8 ± 1.1 W.m-2 for the 1257 Samalas eruption).

- L 333-334: Add commas to help flow: Volcanic injections with known eruption match, and thus better constrained latitude and altitude, have smaller forcing uncertainties...

  Thanks for this suggestion.

- L 395: Wording is very clunky with the double negation, please clarify.

  We modify the text by "when compared to reconstructions, most of the model-reconstruction disagreements can be explained by uncertainties"

- L 487-490, L516-518: Strongly overemphasises the conclusions for NH climate here. Need better quantification/justification why the agreement is better, otherwise please caveat results.

  We have nuanced our results (e.g. "good agreement" instead of "excellent agreement") and the potential limitations highlighted with our simple scaling for NH MJJA temperature are now better quantified thanks to your first major point and the additional analyses we provided in response.

- L 519-520: how can discrepancies be explained by internal variability? I really don't see where this has been taken into account.

  See point 4 in major remarks.

- Fig 1: I like the figure, especially fig. 1f which is a nice visualisation of eruption parameters. However I'd recommend reordering and start with 1f, which is (i) your most important figure and (ii) mentioned first in the text.

Thanks for suggesting that! Done.

- 4b shows GMST for single forcing runs- does it show the ensemble mean for volcanic forcing? And just one implementation of land use. Can you clarify the choice?

  For the volcanic forcing, we show the temperature response to the ensemble mean ERF. For the land use forcing, we only show the temperature response to HYDE 3.3 as the figure was already heavy and difficult to read. As explained in section 2.2, HYDE 3.3 is our "default" land use forcing because (i) it is an intermediate forcing between HYDE 3.2 and KK10; (ii) it is more recent than the other two.

  We modified the figure caption into "FaIR is run using a single set of model parameters. The volcanic forcing corresponds to the 1000-member ensemble mean ERF."

- 5: it is striking that the model here show an ongoing warming trend and a lot more variability at multi-millennial scale than the proxy based data. What are the reasons for this? If Kaufman and Broadman 2023 suggest this could be from volcanic forcing and your results rebut this then it would be worth putting more emphasis on this.

  As pointed out in several papers, volcanic eruptions are the key climate drivers of climate on seasonal-to-multi-centennial timescales (Sigl et al., 2015; PAGES2k-2019; Van Dijk et al. 2024). Thus, we must expect that all model simulations excluding volcanoes (e.g. TRACE-21k) and proxy compilations and data assimilation products with low age resolution and precision over the Holocene (e.g. Kaufman et al., 2020, Erb et al., 2022, Osman et al., 2021) typically have smaller variability than climate simulations with volcanic forcing (as used here, or by Van Dijk et al., 2024). The alleged trend of increased volcanic forcing made by Kaufman and Broadman (2023) derives entirely from their choice of the start and end dates. The sign of the trend would have turned if they had only started their trend analysis at 6,600 yr BP (or anytime earlier) instead of in 6,000 yrs BP. This is evident from the original volcanic forcing reconstruction identifying a "(Mid)-Holocene Active Period" from 9 to 6 ka BP (Sigl et al., 2022) as well as from the millennial-scale volcanic forcing estimated in our simulations (Sect. 5.3). We feel that the reader finds the relevant information on long-term trends of VSSI, SAOD and radiative forcing and climate cooling in this manuscript (in Figs 1 and 3, Table 1) and in previous publications (Sigl et al., 2022; Van Dijk et al., 2024; Figure A1 below).

[Figure]

Figure A1: Simulated NH (0-90°N) temperature (Van Dijk et al., 2024), reconstructed (0-90°N) temperature (Kaufman et al., 2020) and major (>8 TgS) volcanic eruption dates (Sigl et al., 2020).

We added some quantification of this lack of long term trend in the volcanic cooling in the manuscript: "We obtain a difference in volcanic cooling of -0.0005 ± 0.0116 K between the last millennium and mid-Holocene, using the same periods as Kaufman and Broadman, suggesting a small contribution to the 0.1 K cooling visible in the Holocene data assimilation."

---

## Author Comment (AC2)

Verkerk and coauthors combine reduced-complexity volcanic aerosol (EVA_H) and climate (FaIR) models to simulate the global mean surface temperature (GMST) response to volcanic eruptions over the last 9,000 years (6755 BCE to 1900 CE).

To assess the robustness of their simulations, the authors compare their estimates for the 14 largest eruptions between 1250 CE and 1900 CE with numerous climate reconstructions (Schneider et al., 2015; Wilson et al., 2016; Guillet et al., 2017; Pages2k, 2019; King et al., 2021). The discrepancies between the new simulations and climate reconstructions are notably smaller than in previous studies.

The authors address an important topic. The paper is well-written, well-structured, and easy to follow. The figures are clear and informative. And the authors have made all their simulations publicly available.

The methodology section summarizes well the approach taken by the authors, including the forcing datasets used for the new simulations, the paleo-reconstructions and the climate simulations employed to compare the new results.

Additionally, they acknowledge the limitations of their approach, particularly the *Holocene temperature conundrum*, which is also apparent in their ensemble simulations of Holocene temperatures.

The authors emphasize the need for future products based on reduced-complexity models to include seasonal and regional outputs, which would be highly valuable for the paleo community.

I appreciated reading the manuscript and, overall, have very few comments to offer. I recommend the paper for acceptance, as I think the new product provided by the authors represents a valuable resource for the paleo community studying past volcanic eruptions. However, I do have one minor suggestion for the authors to consider.

**Main text:**

- **Comparing simulations with instrumental data**: Pushing the simulations beyond 1900 CE would have been a great addition. Extending the simulations into the 20th century would allow direct comparisons with instrumental data for eruptions such as the 1902 (Santa María), 1912 (Katmai/Novarupta), 1963 (Agung), and 1991 (Pinatubo) events. They could help validate the accuracy of the simulations.

Have the authors considered the possibility of comparing the accuracy of their simulations not only against climate/data assimilation reconstructions but also against instrumental datasets, such as the Berkeley Earth Surface Temperature (BEST) dataset? The BEST dataset offers two products that might be of interest: one estimating GMST since 1850 and another providing annual temperature estimates since 1750 (land-only).

Using these datasets could allow the authors to compare their simulations for the 1815 Tambora, 1831 Zavaritskii (Hutchison et al., 2024), and 1883 Krakatau events with "real" temperature observations. Additionally, the Laki eruption might also be investigated, assuming the instrumental records used by BEST are sufficiently dense to represent a reliable global average (which I am not entirely certain about).

Thank you for your very supportive comments and for suggesting these additional analyses. We have now conducted them and added them to our manuscript. Since the Berkeley Earth dataset has a very sparse spatial coverage before the 1830s (fig 1), we choose to use only the GMST dataset, starting in 1850.

[Figure]

*Figure 1: coverage of the Berkeley Earth dataset in July of the year following a major eruption (a. Laki 1783, b. 1809, c. Tambora 1815, d. Zavaritskii 1831, e. Cosiguina 1835). Horizontal lines delimit the 40-75°N latitudinal band. Temperature are expressed as anomaly with respect to the 1951-1980 mean.*

For comparison, we also included the Cowtan and Way dataset. We generate a 1000-member ensemble for the period 1850-2021 following the same methodology as our 6755 BCE – 1900 CE ensemble. The results are summarized in the text below that has been added to the manuscript.

**Sect. 2.1:**
"In addition to this 6755 BCE – 1900 CE volcanic emission dataset, we also use the CMIP7 volcanic emission dataset to construct an ensemble for the period 1850 – 2021 CE. The eVolv2k dataset provides emissions for the period 1850-1900, emissions

between 1901 and 1978 come from the bipolar ice core record of Sigl et al. (2015). Between 1979 and 2021, we use the satellite record MSVOLSO2L4 (Carn, 2024). For unidentified eruptions or when the injection height is unknown, we apply the same principles as for the 6755 BCE – 1900 CE emission dataset."

**Sect. 2.2:**

**"Forcings for the 1850-2021 CE ensemble**

For the simulations covering exclusively the historical period, we include a larger range of anthropogenic forcings. These include short lived climate forcer (seven species, e.g. black carbon, carbon monoxide), halogen gases (18 species, e.g. chlorofluorocarbons), fluorinated greenhouse gases (23 species, e.g. Hydrofluorocarbons), $CO_2$, $N_2O$ and $CH_4$ emissions, and solar and land-use forcings. Ozone and anthropogenic aerosol forcings are calculated from their precursor emissions."

**Sect. 2.5: Historical observations**

"For the historical period (i.e. after 1850), we compare our 1850-2021 CE ensemble to instrumental observations. We use two observation datasets, the 1850 – present Berkely Earth temperature record (Rohde and Hausfather, 2020) and the 1850 – 2017 Cowtan and Way record (Cowtan and Way, 2014). Both datasets contain monthly temperatures, with a coarser spatial resolution for Cowtan and Way (5° by 5° grid, whereas Berkeley Earth has a spatial resolution of 1° by 1°). The surface temperature is interpolated in region with no station coverage, with Berkeley Earth using a larger number of land stations than Cowtan and Way (around four times more). Here, we use the global annual mean from these spatially resolved datasets to compare it to our simulations."

**Sect 4.2.4: Historical period variability (1850-2021 CE)**

"At a multi-decadal timescale, we observe that the global warming trend in our 1850-2021 ensemble of simulations follow closely the trend in the observations (+0.99 K in Cowtan and Way for the 2010-2016 period relative to the 1850-1900 mean, +1.07 K in Berkeley Earth, +0.97 ± 0.19 for our simulations). Most of the observed annual mean temperatures are within the 95% confidence interval of our simulations (92% of the Berkeley Earth dataset, 95% of the Cowtan and Way dataset). We note that the mid-1930s to early 2000s temperatures appear warmer in the observations.

The observation datasets show a strong interannual variability, with an amplitude similar to the response to volcanic eruptions (e.g. -0.22 K between 1991 and 1992 following the Pinatubo eruption and -0.24 K between 1998 and 1999). To compare the response to volcanic forcing between observations and simulations, we perform a superposed epoch analysis over 6 eruptions that injected more than 7 Tg of $SO_2$ in the stratosphere (Kie Besi 1861, Krakatau 1883, Novarupta 1912, Agung 1963, El Chichon

1982, Pinatubo 1991). We obtain a peak cooling of 0.15 ± 0.06 K on average in our simulations, against 0.12 K in the two observation datasets (Fig. 8.c)."

We also add the following figure (Fig 8 in the manuscript):

[Figure]

*Figure 2: a. GMST for 1850-2021 from our simulations, compared to observations from Berkeley Earth and Way datasets; b. GMST response to the Krakatau 1883 eruption; c. Superposed epoch analysis of the GMST response to 6 eruptions injecting more than 7 Tg of SO2 that occurred between 1850 and 2021 (Kie Besi 1861, Krakatau 1883, Novarupta 1912, Agung 1963, El Chichon 1982, Pinatubo 1991).*

- Line 130: Change Hutchison et al., in review to Hutchison et al., 2024

Thank you, we have updated it.

**Supplementary Material**

- Line 60: "Table S3: Integrated response of the superposed epoch analysis (Error! Reference source not found.d)." There appears to be a reference issue here that should be corrected.

Thanks for pointing that, it now refers to Fig 7 in the main text.